# The Analysis of the Effects of a Fare Free Public Transport Travel Demand Based on E-Ticketing

**Danijel Hojski [1,\*]** , **David Hazemali [2]** **and Marjan Lep [3]**

1   Imovation, d.o.o, Jarška Cesta 10a, 1000 Ljubljana, Slovenia
2   Faculty of Arts, University of Maribor, Koroška Cesta 160, 2000 Maribor, Slovenia; david.hazemali1@um.si
3   Faculty of Civil Engineering, Transportation Engineering and Architecture, University of Maribor, Smetanova Ulica 17, 2000 Maribor, Slovenia; marjan.lep@um.si
\*   Correspondence: danijel.hojski@student.um.si

**Abstract:** The traditional approach in public transport planning was to collect travel demand data for a more extended period and compose timetables to serve this demand. There are two significant identifiable issues. In the rural areas and off-peak hours, public transport operators provide much more capacities than needed. On the other hand, more capacities than scheduled are needed on certain lines at certain departures on some sporadically occurring occasions. The problem is how to react to short-term changes (daily) triggered by exceptional circumstances and events and midterm changes (weekly, monthly basis) in travel demand. We can trigger changes in travel demand chiefly by introducing a desirable (almost for free) tariff system applied to specific populations. No long-term travel response data exists for this kind of intervention, but an immediate response in public transport supply is needed. In Slovenia, public transport for free for the whole population over 65 years was introduced. With the modern ticketing system, which was designed to be as simple as possible for users (that means »check-in only« at the moment of boarding), the research task was to analyze the travel behavior of the retired population, faced with a new attractive option to travel, based on data of purchased tickets and their afterward validation, for better mid-and long-term planning. Our study finds that ITS technology (in this case, e-ticketing system) can satisfactorily solve the discussed planning and management task.

**Keywords:** fare-free public transport policy; smart card data collecting; elderly population mobility; travel demand

## 1. Introduction

In the age of technological progress and digitalization of data at virtually all levels of human activity, the digitalization of data using ITS (Intelligent Transport Systems) technology in public passenger transport (PT) has reached a high level of implementation in most developed countries [1]. Smart card automatic fare collection (SCAFC) and Automatic vehicle location (AVL) enable a standardized procedure for the flow of information, ordered set of data, and automated recording of activities in public transport in the back-office system [1–4]. It is a potent and reliable tool for performing various queries and analyses for transit planners, considering the given data constraints [1]. ITS technology is also implemented in the Integrated Public Passenger Transport System (IJPP) in the Republic of Slovenia. The IJPP system is a national project established in September 2016, representing a unified public transport system in the Republic of Slovenia. The IJPP system includes all operators that provide public transport services in regional public transport. The main principle of the IJPP system is to ensure a uniform and open public transport system in the Republic of Slovenia and to simplify and unify regional public transport for users by introducing a single ticket, which allows public transport under the same tariff conditions regardless of carrier or modality.

On 1 July 2020, the "IJPP free regional ticket or IJPP free ticket Slovenia" (free ticket) for pensioners was introduced in the IJPP system. A free ticket is a unified ticket used in the regional bus and railway traffic throughout Slovenia. The introduction of the free product led to a significant instantaneous change in the IJPP system in the supply and consequently in demand for public transport, for which there is practically no preliminary information for forecasting and planning service capacities. There were no preliminary data to predict the response and characteristics of the free ticket use. The purpose of the study was to monitor the effects on travel demand and travel behavior of target group passengers in regional bus and rail transport. Based on the given data and innovative approaches, a series of analyses were performed, with which we can better understand the effects of the introduction of a free product—the free ticket—for a population group. In this study, the following questions were answered by performing a series of analyses of data obtained from the back-office IJPP system:

- How many of the target population responded to the measure and obtained a ticket, how many used it, and the extent to which the product has been used?
- Are there differences in response and use between the urban and rural environment and by statistical regions of the Republic of Slovenia (regions)?
- What were the day of the week and time of the day choices, and what was the trend of use during the selected period of the research?
- Are the demand patterns for free tickets different from the demand of other users, and what are the differences, if any?

In the remainder of this paper, Section 2 presents an overview of findings in the literature in the field of travel planning in terms of SCAFC technology, introduction, and effects of FFPT cases, and travel behavior characteristics for older populations. Section 2 also highlights literature limitations. Section 3 shows the key characteristics of the introduction of a free product in the IJPP system, and the research methodology. Section 4 presents the results of the analysis and findings of the study, i.e., response, effects, and deviations in traffic demand. Section 5 encompasses a discussion on the findings and recommendations for further analysis.

## 2. Literature Review

Several public transport systems in developed cities use different ITS smart data collection technology implementations [3–5]. A two-way NFC (i.e., near field communication) RFID (i.e., radiofrequency identification technology) compatible communication [6] between the card and the terminal enables a comprehensive data set of recording activities in the back-office system while providing utility maximization for public transport system users. In the back-office system, every activity or transaction carried out is recorded in real-time, enabling an up-to-date overview of events for any period. The back-office system as a consistent archive of real-time activity data for travel demand planning represents untapped potential since, as an IT solution, it is used only in terms of data recording [3,7–11]. Detailed and immediate insight into data on activities and travel demand for public transport enables the implementation of a series of analyses to monitor known parameters and identify changes. Effective travel planning is based on two starting points for implementing activities. Strategic or long-term planning activities include anticipating medium-term and long-term demand due to changed supply. Operational or short-term planning activities are based on the immediate input information, which provides public transport operators with rapid detection and ad-hoc action in the offer of the service or its implementation [3]. From the point of view of travel planning, the back-office system must become easily usable for strategic, tactical, and operational planning, i.e., it ought to evolve into a tool of real-time mobility management.

Both technological and non-technological measures in public passenger transport have the fundamental purpose of making public passenger transport more accessible to users. The level of mobility depends on both the ability to reach the desired destinations using public passenger transport and the accessibility of its use. The accessibility of public

passenger transport depends on the cost of use, technological principles of operation of devices, and understanding of their use. Depending on the circumstances, capabilities, needs, and constraints vary from group to group [12,13].

An attractive change in the tariff system, which enables a specific or selected group of users to use public transport under significantly cheaper conditions or even free of charge, represents a potentially significant change in travel demand for aspects of travel planning [14]. The introduction of Fare-free public policy in public transport (FFPT) is one such approach where one of the main measures is to bring the possibility of using public transport closer to the beneficiaries [15–17].

The intensity of the impact is influenced by factors such as the limitations of FFPT implementation, the previous development of the use of the PT system, preliminary tariff arrangements, ticket subsidies, and geographical and socio-demographic characteristics for the area of implementation of the measure [16–24].

Implementations of FFPT differ in the duration of the implemented measure (for a definite or indefinite period. The implemented FFPT are formed in different ways:

- Temporally-limited (fares are not charged in specific and regular periods of time)
- Spatially-limited (measure applies to a specific section of the PT network)
- Socially-limited (measure embraces a specific group of users) [16].

Several FFPT schemes were implemented to increase the ridership of underused public transport systems in small or medium-sized cities [18]. Case studies of the introduction of FFPT in public transport, mostly in urban areas, have shown that the measure impacts raising travel demand, with effects varying from city to city. Selected examples of full FFPT in urban and regional PT are shown in Table 1.

The measure of introducing a free ticket in IJPP is an example of a FFPT, which is socially limited to a specific group of users—pensioners. Based on this criterion, comparable examples of introduced FFPT in urban PT include Canberra, Australia (elderly) and Shanghai, China (pensioners), while for the regional PT, a comparable example is that of the UK. The FFPT measure in the United Kingdom is temporary-limited, as it does not apply to morning peak periods (until 9:30 am), while beneficiaries of a free ticket in Slovenia are not limited in time.

In Slovenia, the introduction of FFPT is not a complete novelty, as PT is free in three smaller cities (Jesenice, Nova Gorica, Velenje). Compared to the measure in the study, they differ in the type of traffic (i.e., urban PT), restriction (i.e., unlimited use), and, above all, the size of PT systems and networks.

In Slovenia, the introduction of free ticket represents a unique opportunity to determine the effects on travel demand caused by such an event. The specificity of the studied measure is reflected in the following characteristics:

- type of public transport, i.e., regional public transport,
- the area of implementation of the measure, i.e., at national level (spatially-limited),
- change of offer, i.e., free use of the entire IJPP system without restrictions (temporary-limited),
- specific target group of beneficiaries (socially-limited).

The FFPT measure, introduced in the IJPP system, affected the pensioners. Represented predominantly by the elderlies (65+ years), the pensioners are an inactive population, consisting of the unemployed people and the people not included in educational programs. The inactive population makes a larger share of trips outside peak hours, as the purpose of the trip often allows for greater flexibility in terms of destination and time [25]. Elderly trip-making tends to occur during the midday peak and daylight hours mostly. Most elderly individuals make their trips between 9:30 a.m. and 3 p.m. [26]. They also tend to travel shorter distances and make fewer trips than other adults (25–59 years) [27].

Due to the specific circumstances of the introduction of a free ticket in Slovenia, as described above, it was not possible to draw direct parallels with existing studies.

**Table 1.** Selected examples of full FFPT in urban and regional PT.

| FFPT Form | Key Features | Type of PT | Examples of FFPT Programs |
|---|---|---|---|
| Full, unlimited | A free ticket system implemented in most routes and services within a given urban PT network, which is available ordinarily to the vast majority of its users for at least 12 months. | Urban | Hasselt (Belgium), Aubagne (France), Tallin (Estonia), Frýdek-místek (Czechia), Nova Gorica, Veljenje, Jesenice (all Slovenia) |
| Socially limited | A ticket-free system that is limited in one or more ways. | Urban | Canberra, Australia (elderly), Shanghai, China (pensioners) |
| Temporary and socially limited | Fares are suspended in specific yet regularly occurring periods. | Regional | United Kingdom (elderly) |

## 3. Methodology

### 3.1. Field of Research

The IJPP system uses technology that guarantees users simple, easily understandable, and quick usage. A contactless "Check-In" ticket validation system has been set up to achieve the user-friendly principle. There is a single tariff for the IJPP system, based on which users can use the links to which they are entitled under the same financial conditions as the products on the single IJPP card. Due to the implementation of smart card technology, every transaction within the IJPP system is recorded in real-time. As the card is placed on the terminal, contactless communication is carried out, where every operation, i.e., transaction, is recorded and stored in the back-office.

The IJPP free ticket product was introduced in the IJPP system on 1 July 2020. The IJPP system covers all regular regional bus and railway traffic lines throughout Slovenia. With the measure, the population group of pensioners (beneficiaries) are entitled to the product of a free ticket. Pensioners are persons who receive an old-age, disability, or survivor's pension under the regulations of the Republic of Slovenia or from a foreign pension insurance institution and are not classified as employed, unemployed, or students according to their activity status [28].

According to the data for the third quarter of 2020, Slovenia had a population of 2,100,126 [29] at the time of the introduction of the measure and the conduct of the survey. 532,169 persons were entitled to a pension, which was 25.3% of the total population [30]. Demographical and spatial characteristics of Slovenia are shown in Table 2.

**Table 2.** Demographical and spatial characteristics of Slovenia.

| | Surface [Km$^2$] | Population | Number of Beneficiaries |
|---|---|---|---|
| Slovenia | 20,271 | 2,100,126 | 532,169 |
| City municipalities | 1883 | 755,651 | 187,466 |
| Ljubljana | 275 | 295,504 | 67,946 |

As the free ticket measure was introduced, a new user status was granted to beneficiaries for classification. A special card was presented for the free ticket product, which each beneficiary received who applied for a free ticket. The administrative cost of making and initializing the name card was 3 EUR and represented the only cost for the unlimited use of the IJPP system. For every new IJPP card, a unique card identifier (number) is assigned. In addition, each user of the IJPP system is assigned user status and each product a product status.

The research was spatially limited to the area of operation of regional bus and railway PT in the entire territory of Slovenia. Furthermore, the parameter of "statistical region" was added based on the statistical data of the Statistical Office of the Republic of Slovenia (SURS). Regions of Slovenia are shown in Figure 1. Characteristics, relevant for the study, are shown in Table 3.

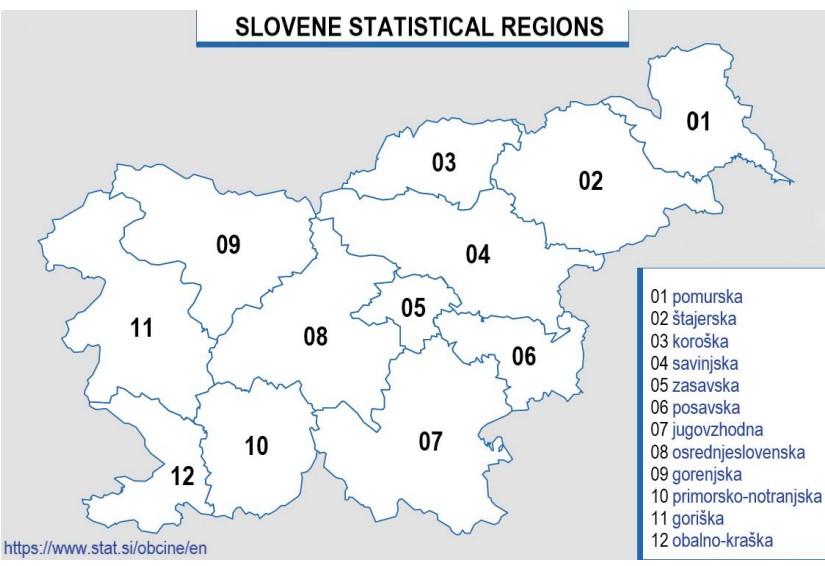

**Figure 1.** Statistical regions of Slovenia (source: SURS).

**Table 3.** Characteristics of statistical regions of Slovenia.

| Statistical Region | Surface Share | Population Share | Share of Beneficiaries | Average Gross Pension [Eur] | GDP by Region Per Person [Eur] |
|---|---|---|---|---|---|
| Gorenjska | 10.5% | 9.8% | 26.4% | 70,774 | 19,833 |
| Goriška | 11.4% | 5.8% | 28.6% | 69,937 | 19,930 |
| Jugovzhodna | 13.3% | 7.0% | 25.3% | 65,982 | 21,630 |
| Koroška | 5.1% | 3.5% | 27.2% | 67,284 | 17,885 |
| Obalno-kraška | 5.2% | 5.6% | 26.3% | 73,283 | 22,627 |
| Osrednjeslovenska | 11.1% | 25.3% | 24.3% | 80,305 | 31,169 |
| Podravska | 10.7% | 15.8% | 25.7% | 65,430 | 17,838 |
| Pomurska | 6.6% | 5.7% | 27.0% | 58,289 | 14,937 |
| Posavska | 4.8% | 3.7% | 26.8% | 64,950 | 18,314 |
| Primorsko-notranjska | 7.2% | 2.5% | 27.4% | 66,658 | 15,837 |
| Savinjska | 11.7% | 12.5% | 26.0% | 68,781 | 19,987 |
| Zasavska | 2.4% | 2.8% | 28.3% | 71,198 | 11,574 |
| SLOVENIA | | | 25.3% | 70,950 | 22,083 |

The research was limited in time to the validity period of introducing a free ticket for the following four months: July, August, September, and October 2020. In mid-November, public transport in Slovenia was stopped due to the declaration of a COVID-19 epidemic, and research was not possible. The suspension of public transport followed several previous restrictive measures due to the deteriorating epidemiological situation in October, which clearly shows a decline in traffic compared to the same period in 2019. In July and August, classes and lectures in educational institutions were generally not held due to the summer vacation. The use of IJPP was thus significantly lower during this period. As a result, the data used for more detailed analyses of the application of the IJPP were covered for the first two weeks of September 2020. Travel demand was also affected by the weather conditions, which were very favourable for this part of the year, with temperatures up to 33 °C, and no major precipitation recorded [31].

### 3.2. Data Collection

Data set

The IJPP operates at 5417 bus and railway stops and 2883 lines running in the system. In 2020 there were approximately 500,000 registered users, which provided some 12,000,000 validations. Every bus and train under IJPP is equipped with a terminal that

contains a reader with an integrated GPS unit. Consequently, the data set used in this study came from the SCAFC and AVL systems of the IJPP. All validations contain the information about the transaction location (bus/train stop ID) and time (time and date).

The research included data on types of purchase transactions, activation, and validation of tickets. Every ticket issue transaction is recorded as a purchase. The first use of the product is recorded in the IJPP system as product activation, while each subsequent use of the product is recorded as product validation. Activation thus represents the first validation of the product. Part of transactions scheme of the IJPP system, relevant for the study, is shown in the Figure 2.

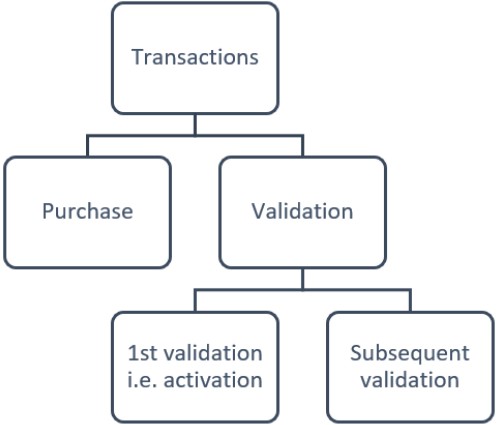

**Figure 2.** The transactions scheme of the system IJPP.

Data mining

The SQL (Structured Query Language) programming language was used to obtain data for the study. Based on the SQL, several queries were performed. Further, the data were stratified by filtering by parameters:

- at the product type level, by the product status,
- at the individual card level, by the unique identifier.

To conduct the analysis, all successful transactions were covered. A 100% data sample was obtained for the research period. For the study, the aggregated IJPP data set was used. A card identification number was used to separate the products. For the analyses, the following data were included for each transaction:

- unique card identifier,
- type of transaction,
- information on user status and product type,
- time and date,
- unique identifiers for the terminal,
- unique identifier for stop, timetable, and ride, and
- municipality and settlement.

Based on further data processing, the authors included data on the Republic of Slovenia's post office and statistical region for each transaction, where the product was issued or validated. There were 5,148,708 transactions covered in the 4-month surveyed period in 2020. Defective or inappropriate transactions, which accounted for less than 0.1% of all transactions, were excluded from the Data Set. Any transaction with a value of "NULL" or without a value for at least one of the parameters was identified as inadequate.

## 4. Results

### 4.1. Response to the Free Ticket Introduction

In the period of four months after introducing the free ticket, 148,766 beneficiaries obtained the card and the product, representing 28% of the total population eligible for

the free ticket. The share of holders among beneficiaries varied considerably from region to region, ranging from 14% to 40%. A higher share of free ticket holders was recorded in regions with higher average pensions. These regions also have a higher share of urban areas and are more developed in tourism. The shares of free ticket holders in comparison with beneficiaries by region are shown in Figure 3.

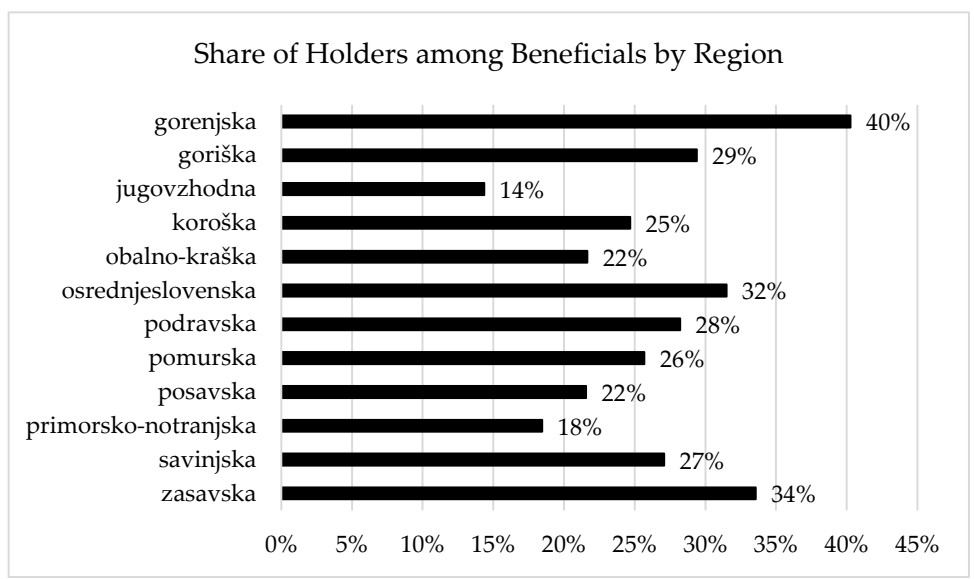

**Figure 3.** Share of issued free tickets in comparison with beneficiaries by region.

The authors determined how the issuance of free tickets took place for the research period by weeks. The response of beneficiaries was the highest in July and amounted to as much as 62.5% of all purchases in the research period. The issuance of a free ticket has not yet meant the actual use of the IJPP system. The activation data determined the share of free tickets used. During the study period, 66,669 were activated, which means that only 45% of all issued tickets were used. The main features of the response to the introduction of a free ticket are shown in Table 4.

**Table 4.** The main features of the response to the introduction of a free ticket.

| The Main Features of the Response to the Introduction of a Free Ticket | |
| --- | --- |
| Number of issued tickets | 148,766 |
| Response among beneficiaries | 28% |
| Share of activated tickets | 45% |
| The average usage of a single ticket | 11 times |

### 4.2. Use of Free Ticket

The responsiveness of the obtained free ticket activation is recorded as the reaction time between the types of purchase and activation transactions. The average reaction time was 22–75 days. The authors analyzed the frequency of use to understand the use of the free ticket product. Further, they determined the frequency of use by comparing all activations and validations for all activated products. The average frequency of use for the study period was 11 validations. Frequency of free ticket validations is Figure 4.

The results show that the free ticket was primarily not used regularly or daily. The authors found that as many as 33% of all tickets were only used once or twice. Ten or fewer validations were thus recorded for 75% of all activated tickets.

The trend of validations differs from the trend of ticket purchases. Just over 28% of all validations were recorded in the first two months. The curve then increased over the weeks, peaking in the September, when 35.5% of all validations over the research period were

recorded. The measure of introducing a free ticket at the end of the four months research period has seen a sharp decline in beneficiaries' response, which is evident from purchases. The trend of use of the acquired tickets, which is evident from validations, also indicates a decrease. Trend of use of the acquired tickets, which is evident from validations, indicates a declination in the use as well. The reasons for the recorded negative trend in demand for free of charge could not be determined from the surveyed data. The deterioration of the epidemiological situation in Slovenia due to COVID-19 and the end of the summer tourist season may have contributed to the negative trend in October. The trend of purchases, activations, and validations of free tickets is shown in Figure 5.

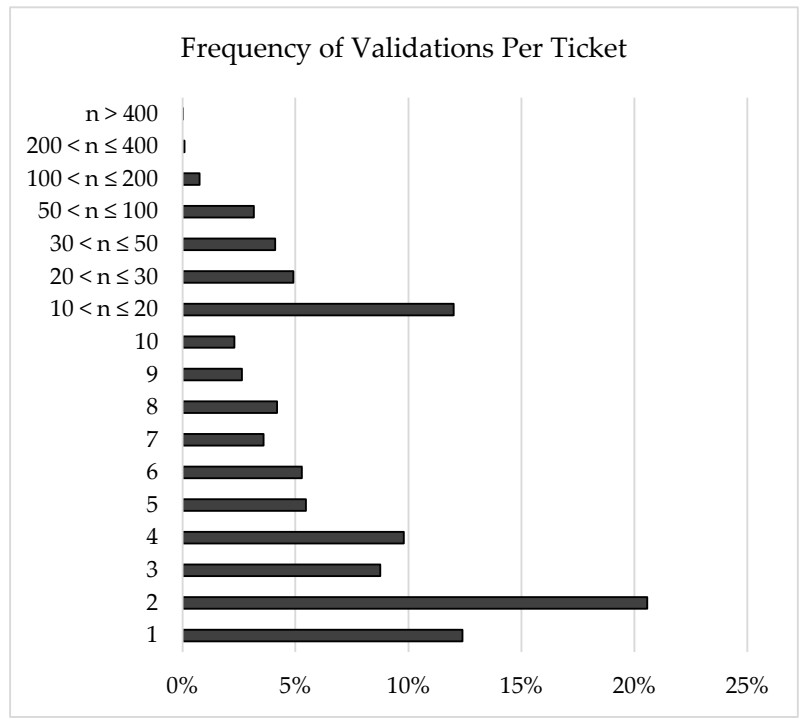

**Figure 4.** Frequency of free ticket validations.

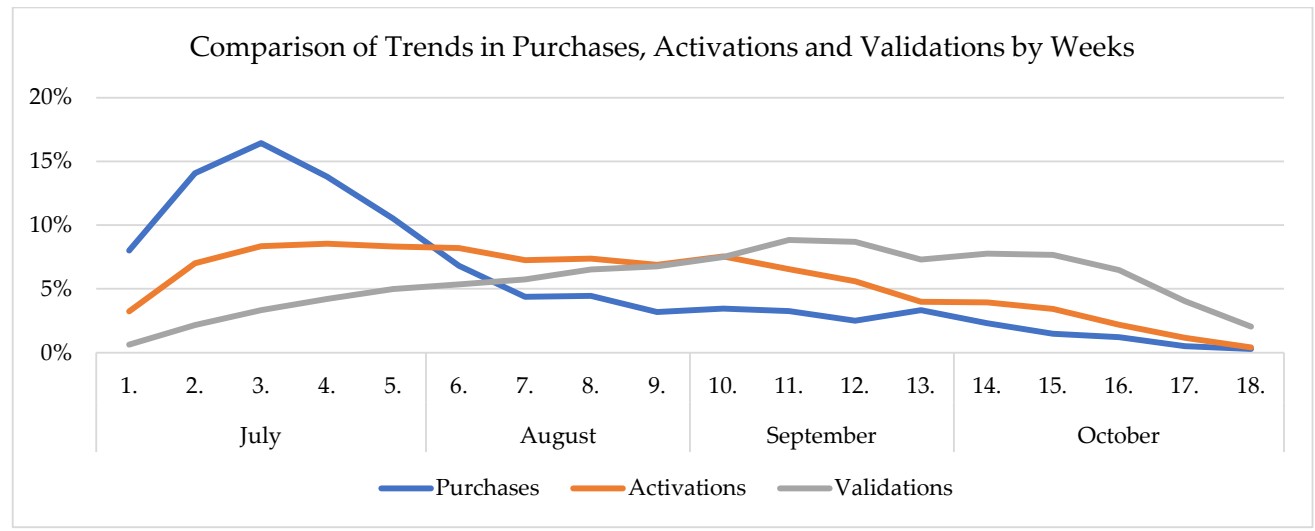

**Figure 5.** Comparison of trends in purchases, activations, and validations of free ticket by weeks.

The spatial distribution of validations was performed at the level of statistical regions of the Republic of Slovenia. The authors found the validations not to be evenly distributed

among the regions. To obtain representative usage data, validations with free tickets were compared with the share of beneficiaries who obtained a free ticket. The number of issued free tickets, shares of free ticket holders' and shares of validations of the free tickets by region are shown in Table 5.

**Table 5.** Validations of free tickets by region depending on the holders' shares among the beneficiaries.

| Statistical Region | Issued Free Tickets | Holders among Beneficiaries | Validations | Validations Per Issued Ticket |
|---|---|---|---|---|
| Gorenjska | 21,450 | 40.3% | 124,989 | 5.8 |
| Goriška | 9940 | 29.4% | 28,537 | 2.9 |
| Jugovzhodna Slovenija | 5216 | 14.4% | 25,502 | 4.9 |
| Koroška | 4826 | 24.7% | 20,633 | 4.3 |
| Obalno-kraška | 6523 | 21.7% | 67,393 | 10.3 |
| Osrednjeslovenska | 39,816 | 31.5% | 175,936 | 4.4 |
| Podravska | 23,526 | 28.2% | 89,132 | 3.8 |
| Pomurska | 8185 | 25.7% | 31,023 | 3.8 |
| Posavska | 4366 | 21.6% | 18,454 | 4.2 |
| Primorsko-notranjska | 2609 | 18.5% | 10,070 | 3.9 |
| Savinjska | 18,042 | 27.1% | 84,666 | 4.7 |
| Zasavska | 5536 | 33.6% | 55,562 | 10.0 |

The most validations were recorded in the Osrednjeslovenska region (175,936), followed by the Gorenjska region (124,989). Other regions had a significantly lower number of validations, under 100,000. To obtain representative data, the number of validations was compared with the number of issued free tickets. The number of validations per issued ticket was obtained from the comparison. The free ticket was most used in the Obalno-kraška (10.3) and Zasavska regions (10.0). A higher share of use was also recorded in the Gorenjska region. Higher shares of validations, except for the Zasavje region, were again recorded in more developed regions, especially those with tourism. In the Zasavje region, the share may be high due to less developed road connections, especially with the Osrednjeslovenska region, where the capital city Ljubljana is located.

*4.3. Travel Behavior of Free Ticket Users*

The study analyzed data for the time of the day choice and the time of the week choice. Most free ticket validations were performed on weekdays (88%). Travel demand of free ticket holders by day of the week is shown in Figure 6.

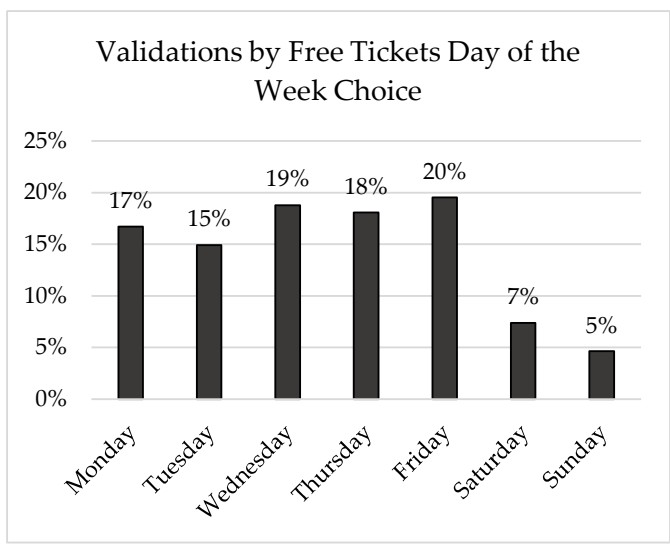

**Figure 6.** Validation by free tickets by day of the week choice.

The authors identified the time of the day choice distribution on validations based on the analysed validations of free tickets by hours. In the analysis, the morning and early afternoon hours (9–15) stand out when 75% of all validations with free ticket were performed. The distribution by part of the day differs for weekdays and weekends. On weekends, especially on Sundays, an increase was observed during the afternoon and evening hours (15–20); it amounted to 50% of all daily validations. A smaller number of validations were recorded for noon hours (11–14). The trend of validations for weekdays, Saturdays, and Sundays by time of the day choice is shown in Figure 7.

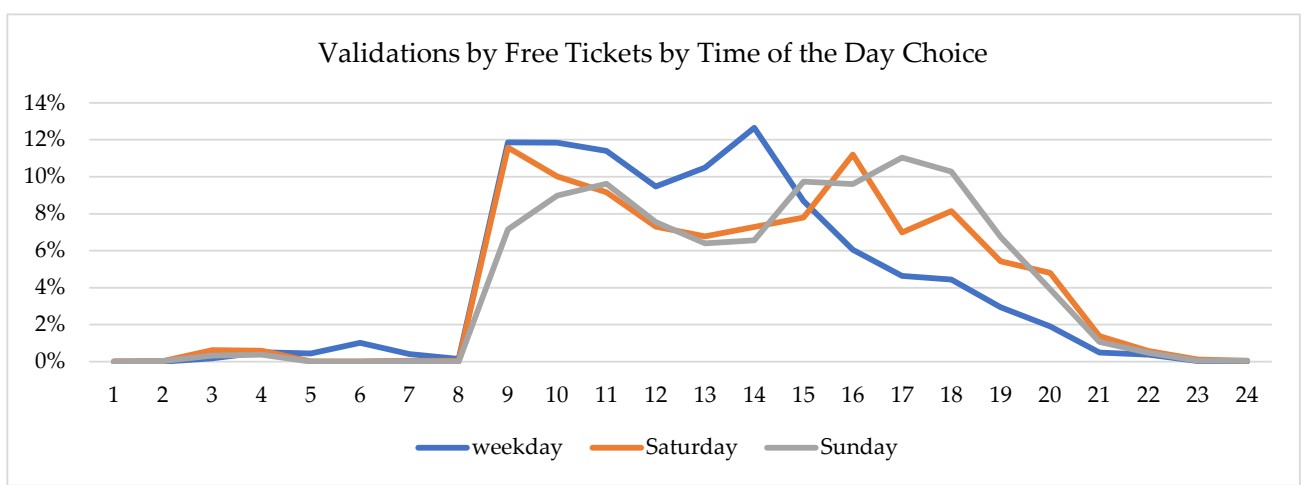

**Figure 7.** Time of the day choice for weekdays, Saturdays, and Sundays.

The time of the day distribution of validations by region is not evenly distributed. The spatial distribution of validations by the time of the day choice is shown for the most attractive tourist regions (Gorenjska, Obalno-kraška, and Osrednjeslovenska). These regions accounted for 62% of all tourist visits in Slovenia in 2019 [32]. During the weekend, an increase in the number of validations in was observed in both peak periods. During the afternoon peak period, the shares of validations in these regions are significantly higher. The Gorenjska and Obalno-kraška regions thus account for 45% of all validations in the afternoon peak period. It can be inferred from the results that a large proportion of trips were for vacation purposes.

### 4.4. Effects of the Introduction of Free Tickets on the IJPP System

The number of pensioners who used the IJPP system before the introduction of the free ticket (i.e., between 2016 and June 2020) could not be determined, because there was no product status "pensioner" in the IJPP ticketing system. Consequently, prior data on their use could not have been obtained exactly. Prior to the introduction of the free ticket, there were two product statuses in the IJPP system, i.e., regular ticket and subsidized ticket for the population included in the education programs. Validations with the subsidized ticket accounted for the vast majority of all validations, namely 96%. Pensioners who had previously used the IJPP system must use regular ticket. By comparing the shares of purchases of IJPP tickets for the investigated period of 4 months (1 July–1 November) for 2019 and 2020, the share of sales of regular tickets in 2020 did not decrease. It can be assumed that the percentage of pensioners who previously used the IJPP system was significantly less than 4%.

Some data for before-after analyses are available for Municipality of Ljubljana (MOL) and Ljubljana Urban Region (LUR) [33]. In 2014 household surveys, travel diaries on the sample of 1% of the entire population were collected. The data showed that pensioners in LUR accounted for 10.7% of public transport users in the region. The entire PT, both urban and regional, was covered in the research. MOL and LUR PT user shares by social status are shown in Figure 8.

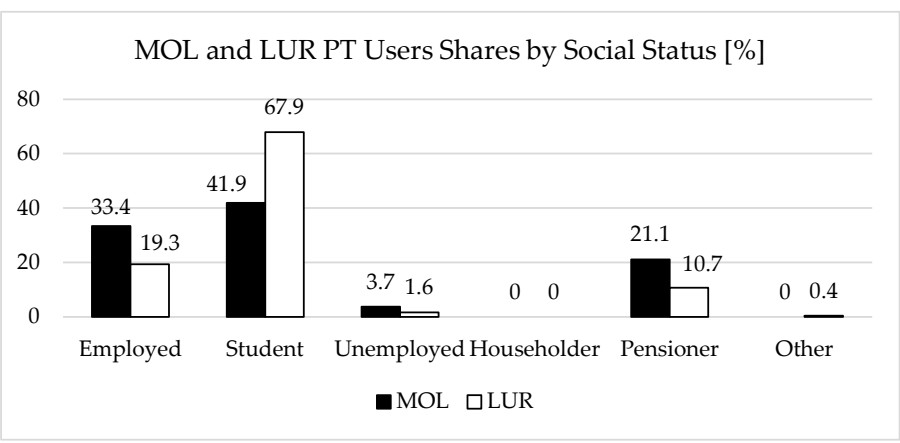

**Figure 8.** MOL and LUR PT user shares by social status.

The authors compared the findings on the travel patterns of free ticket users with the characteristics of other users in the IJPP system (non-free tickets validations) to evaluate the induced travel demand caused by free tickets. In the first two weeks of September 2020, 1,008,584 validations were performed, of which 90,928 were free tickets, representing 9.0% of all validations. By day of the week choice 97% of non-free ticket validations were carried out weekdays, while the share among free tickets was lower (88%). Therefore, the contribution from free tickets validations to the full use of the IJPP system during the weekend was 28.8%.

The next indicator of characteristics was to analyze the time of the day choice for free and non-free tickets. Data for weekdays and weekends were analyzed. It was found that the time of the day choice for free tickets differs significantly from non-free tickets validations. Discrepancies were found on both weekdays and weekends. The time of the day choice with a free ticket does not differ significantly for weekdays and weekends. The vast majority of trips, i.e., more than 85%, were done between 9 and 6 p.m. On the other hand, major discrepancies were found in the daily flow of non-free ticket validations for weekdays and weekends. For non-free ticket validations, there were significant peak periods during the workdays, i.e., morning (6–8) and afternoon (13–16) peaks, where a total of just under 70% of all validations were recorded. During the weekends, most validations (just under 80%) were performed in the afternoon and evening (15–21). A comparison of the daily flow of free ticket and non-free tickets validations in the IJPP system on workdays are shown in Figures 9 and 10. A comparison of the daily flow of free ticket and non-free tickets validations in the IJPP system on weekends are shown in Figures 11 and 12.

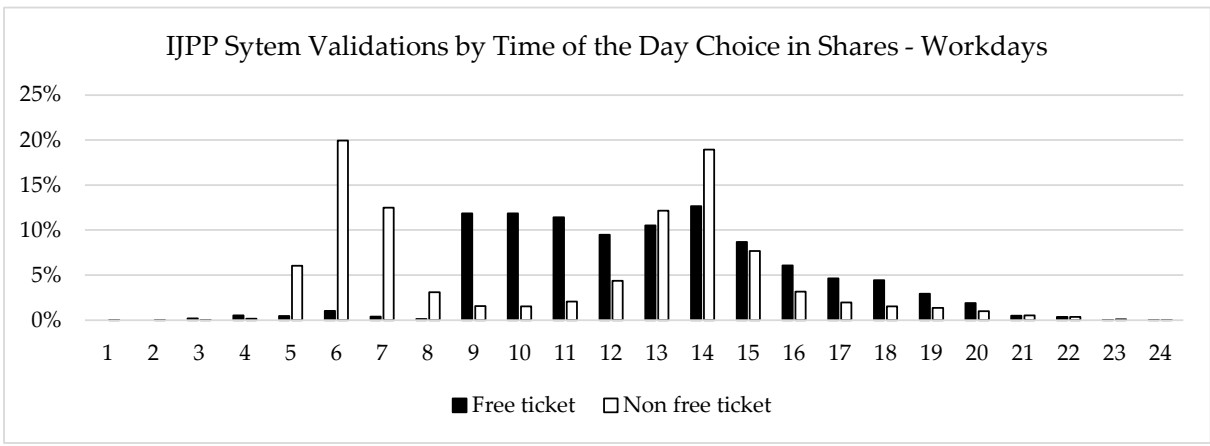

**Figure 9.** Time of the day choice comparison of free ticket and non-free tickets validations in the IJPP by shares—workdays.

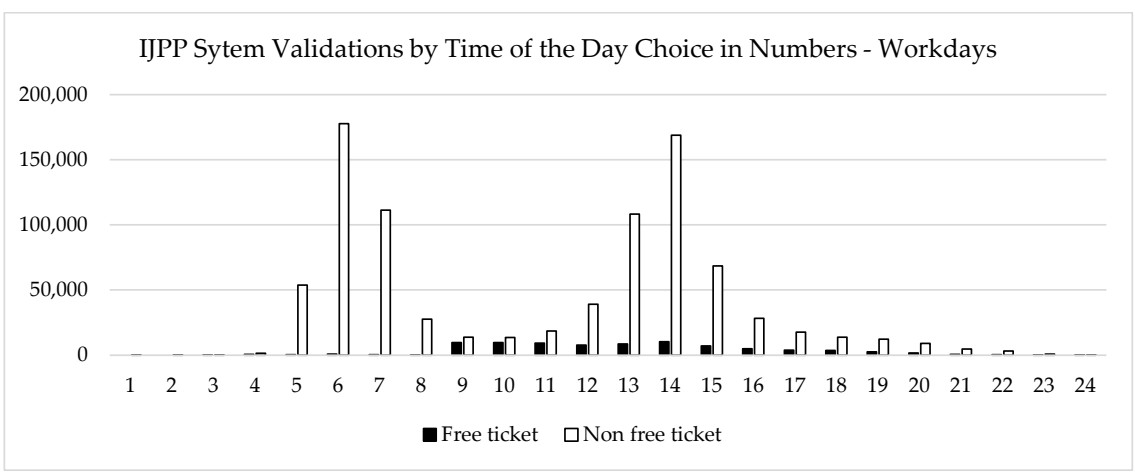

**Figure 10.** Time of the day choice comparison of free and non-free tickets validations in the IJPP by absolute numbers—workdays.

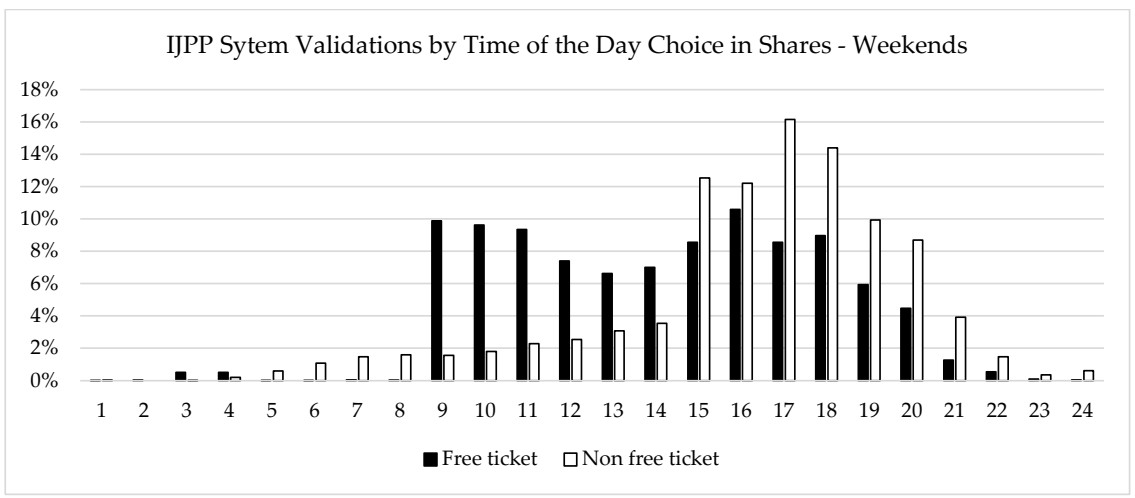

**Figure 11.** Time of the day choice comparison of free ticket and non-free tickets validations in the IJPP by shares—weekends.

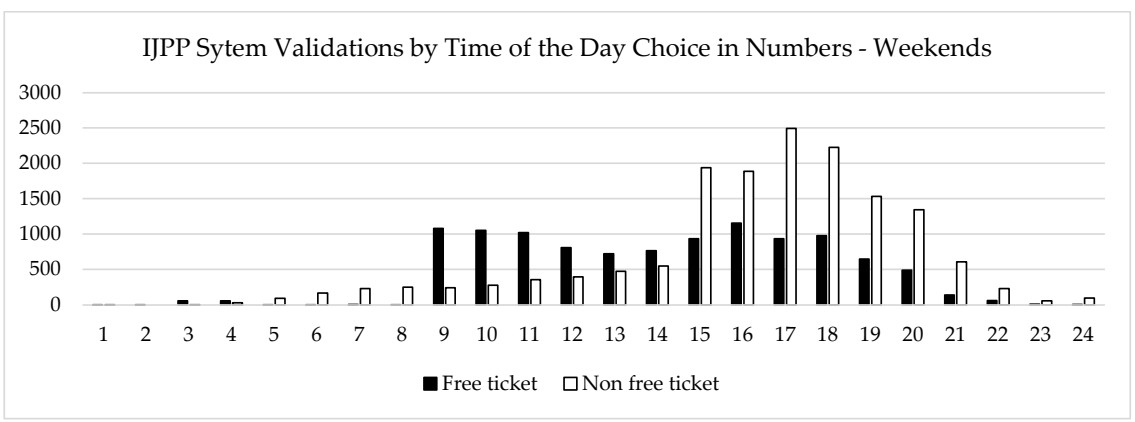

**Figure 12.** Time of the day choice comparison of free and non-free tickets validations in the IJPP by absolute numbers—weekends.

The most pronounced deviation between free and non-free ticket use for workdays was in the first part of the day (6–13). In the non-free ticket morning peak period (6–9), the

share with a free ticket is negligible. In the forenoon (9–12), there was a significant increase in free ticket validations and an even greater decline in non-free ticket validations. In that period, validations with a free ticket accounted for just under a third of all validations. The share of use from free ticket validations was again meager in the afternoon and evening.

Due to the small number of non-free ticket validations during the weekend, the contribution of validations with a free ticket was more significant than on weekdays. In the forenoon period (9–12), free ticket validations accounted for more than 75% of all IJPP validations and more than 60% for the first part of the afternoon (12–14). In the second part of the afternoon (15–18), they accounted for a third of all IJPP validations in the measured period.

A comparison of the distribution of validations by part of the day shows that the time of the day choice of free ticket users differs significantly from other IJPP users' traffic flows. Differences in the time of the day choice were determined based on a comparison of validations by region. Significant discrepancies were found in the distribution of validations of free and non-free tickets by region. Both, on weekday and especially on the weekends, significant deviations in the share of validations were found in the Gorenjska (up to 3.6 times higher), Obalno-kraška (up to 5.5 times higher), and Osrednjeslovenska region (up to 2.7 times lower). Deviations in the distribution of validations were recorded in other regions as well, but they were not as pronounced. The compared data shows differences in the distribution of validations by region and between validations in urban and rural areas. Data on urban areas were obtained based on a settlement where validation was carried out. These settlements with the same name as municipalities with the status of urban municipalities were considered urban areas. Among the validations of free tickets, the share of validations with a free ticket in a rural area was 58%, which indicates a significantly higher share of validations from rural areas. For non-free tickets validations, 62% of validations were performed in the urban area. Non-free tickets validations are also characterized by a very pronounced centralization of validations in the capital city of Ljubljana. 43.3% of non-free ticket validations were performed in the urban area of Ljubljana. For comparison, only 17.4% of free ticket validations in urban area of Ljubljana were recorded. A comparison of spatial distribution between free ticket and non-free ticket validations is shown in Table 6. Shares of free ticket validations in terms of rural and urban areas are shown in Figure 13.

**Table 6.** A comparison of spatial distribution between free ticket and non-free ticket validations.

|  | Shares of Validations [%] | |
| --- | --- | --- |
|  | **Free Ticket** | **Non-Free Ticket** |
| Rural area | 62 | 42 |
| Urban area | 38 | 58 |
| Urban area of Ljubljana | 14.4 | 43.3 |

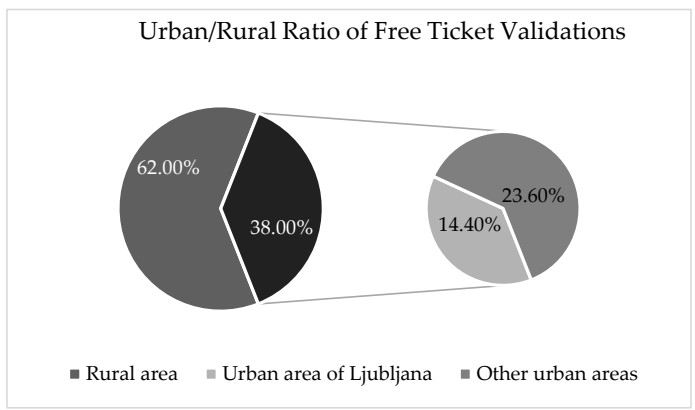

**Figure 13.** Urban/rural ratio of free-ticket validations.

## 5. Conclusions

With the introduction of a free ticket in the IJPP system, there was a significant change in the supply and consequently in the travel demand for public transport for the target group of beneficiaries, representing 25.3% of the 2,100,126 total population in the Republic of Slovenia. The analysis showed that the IJPP system used a minimal number of beneficiaries before introducing the free ticket, as 96% of all validations came from subsidized tickets for the population in educational processes. During the research period, i.e., four months after introducing the free ticket, 148,766 free tickets were issued, representing 28.0% of all eligible persons for the free product. The trend of issuing tickets was the highest in the first month after the introduction, when it presented 62.5% of all tickets from the research period were issued. According to the number of beneficiaries, the distribution of purchases by statistical regions was not uniform. The shares between the regions varied from 14.2–40.0%. The analysis of the actual use of tickets revealed that only 45% of all issued tickets were activated, which shows the psychological moment, where a large part of the population exercises the right to a bonus that they do not intend to use. Most activated tickets were not used daily, as in the four-month research period, the single activated ticket recorded only 11 validations on average.

The most representative two-week research period was chosen to analyze the free ticket use, i.e., the first two weeks of September. Other study periods were significantly affected by summer holidays, restrictive measures, and other influences related to COVID-19 coronavirus disease. During the 4-month study period, a total of 90,928 free ticket validations were recorded during the study period, representing 9.0% of the total use of the IJPP system. Based on a comparison of the number of validations by region, considering the shares of beneficiaries by region that received a free ticket, it was found that the use of free tickets by region was markedly uneven, with usage rates differing by more than ten times. A higher share was recorded in more developed and tourist-attractive regions. An analysis of validations by day of the week, time of the day, and by regions in the Republic of Slovenia was performed to understand travel demand characteristics. Based on a comparison with the travel demand of other users of the IJPP system, the authors found that the travel demand characteristics of free ticket holders differ significantly. Differences were found at all levels of analysis. On a weekly basis, free tickets had a higher share of weekend use (12%) than other IJPP tickets (3%). Discrepancies for the time of the day choice and spatial distribution were found for users of the free ticket compared to other users of the IJPP system.

The time of the day choice for weekdays and weekends did not differ significantly with the free ticket validations. About 85% of all daily validations with a free ticket were performed in the forenoon and afternoon (9–18). On the other hand, significant differences were found between weekdays and weekends for forenoon free ticket validations. For weekdays, there were two distinct peak periods (6–9, 13–16), where 70% of all free ticket validations were performed. Comparing the daily flow of validations with free tickets and non-free tickets showed significant differences both on weekdays and weekends. The highest share of validations with a free ticket was recorded in the morning and early afternoon (9–15), especially during the weekend.

From the spatial point of view, differences were found at the levels of regions and the urban and rural environment. The use of a free ticket is less centralized. 17.4% of validations were performed with a free ticket in the urban area of the capital of Slovenia, Ljubljana. Significantly more non-free tickets were made, namely 43.3%. The compared tickets' validation ratios between urban and rural areas are almost reversed. 42% of validations were performed with a free ticket in urban areas.

The study representatively demonstrated a difference in the characteristics of the travel demand of free ticket users. For travel planning, this means a change in travel demand in a very brief time. The use of modern approaches to ITS technology enables immediate and detailed insight into the data that were processed and analyzed with innovative approaches. The characteristics identified with the introduction of the free ticket may in the future be the

starting point for automated detection of sudden deviations in travel demand of morning and forenoon validations, which would, based on known correlations between traffic flows in the system, result in immediate proposals for afternoon and evening traffic flows. Such an algorithm of automatic demand detection would enable operators to use capacities more efficiently—to reduce delivered overcapacities and to avoid overcrowded busses or not served costumers due to lack of available seats, as a main performance indicator.

**Author Contributions:** Conceptualization, D.H. (Danijel Hojski), M.L. and D.H. (David Hazemali); methodology, D.H. (Danijel Hojski) and M.L.; software D.H. (Danijel Hojski) and D.H. (David Hazemali); validation, D.H. (Danijel Hojski) and M.L.; formal analysis, D.H. (Danijel Hojski); investigation, D.H. (Danijel Hojski), M.L. and D.H. (David Hazemali); resources, D.H. (Danijel Hojski) and D.H. (David Hazemali); data curation, D.H. (Danijel Hojski); writing—original draft preparation, D.H. (Danijel Hojski), M.L. and D.H. (David Hazemali); writing—review and editing, M.L. and D.H. (David Hazemali); visualization, D.H. (Danijel Hojski); supervision, M.L.; project administration, M.L.; funding acquisition, M.L. All authors have read and agreed to the published version of the manuscript.

**Funding:** This research received no external funding.

**Institutional Review Board Statement:** Not applicable.

**Informed Consent Statement:** Not applicable.

**Data Availability Statement:** Data sharing is not applicable to this article.

**Acknowledgments:** Ministry of infrastructure of the Republic of Slovenia, Sustainable mobility and transport policy directorate and SŽ-Potniški promet, d.o.o. for access to the IJPP system data base.

**Conflicts of Interest:** The authors declare no conflict of interest.

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
