# Peer review of "The Analysis of the Effects of a Fare Free Public Transport Travel Demand Based on E-Ticketing"

_sustainability, doi:10.3390/su14105878_

Round 1
Reviewer 1 Report
The title, the abstract, the core text, and the conclusions are incoherent: the title offers a general methodology, while the abstract reports on the travel behavior of the retired population. Moreover, later the Results chapter shows basic statistics from the transaction database, but the Conclusions chapter generalizes the results to the whole population. The analysis lacks any scientific standards.
The first sentence in the Abstract seems to be confusing: on the one hand, the authors state the public transport supply is inefficient. On the other hand, operators provide much more capacities than needed. Such a demand-oriented public transport policy can be the key to decreasing the number of private cars. Sometimes such transport policy may be cheaper for the whole society than suffering from high private car ration in the modal split. Nevertheless, aiming for higher efficiency of the public transport system is undisputedly a critical point.
The methodology chapter lacks several pillars for the research: the research scope needs to be strengthened, there is no methodology review. Although the authors stated the Slovenian case is unique (lines 132-133), they didn't introduce their country's case, and they didn't show up existing studies from the scientific literature. Consequently, the arguments why other studies are incomparable are missing and inadequate. Moreover, the introduction to the dataset seems to be of low quality, e.g., how the card identifier does prove that the holder belongs to the selected groups of users (elderly people)? What are the user numbers and their distribution before the analyzed period? The PT system's current and investigated efficiency levels (and key performance indices) are also missing. Nevertheless, how do the authors define efficiency?
There is a remarkable need for an extensive English language and style revision.
Reviewer 2 Report
The work analyses the impact on the travel demand of the introduction of a free ticket in the Slovenian (long distance) public transport system.
The topic is absolutely relevant for current research in the field. Honestly, I did not know about the transport system organization in Slovenia. In my opinion, the development of an integrated public transport as well as the introduction of free tickets or appealing tariffs are excellent measures to set the travel demand in a sustainable-mobility perspective.
The description of the problem is pretty accurate, the objectives are clear even though the methodological approach appears quite basic. The paper is easy to read, English is fluent and also the references are appropriate.
My main request to the authors is to add a more in-depth discussion (explanation) of some of the presented results. Although definitive conclusions may not be drawn from several of the obtained results, the authors should certainly provide their interpretation of the results based on their knowledge of the transport system and the social, economic, and urban conditions in Slovenia. To provide an example, at lines 266-270 it would be beneficial to offer a possible explanation for the different percentages of use of free tickets in the various regions. Other such cases appear in subsections: 216-217, 252-253, 266-270, 287-294, 331-336, 376-377, 387-388, 394-401.
Minor:
Introduction: besides the long-distance public transport, briefly introducing the transport demand and supply in Slovenia can help the reader understand the general picture.
Results: Are you able to assess whether there was at least a low share of beneficiaries who frequently used public transport even before the introduction of the free tariff policy?
82-84: Consider rephrasing this sentence.
193-213: In order to make the text more intelligible, try to reformulate and synthesize this subsection avoiding repeating the same sentences.
Figure 5: You can make the figure more intelligible by introducing 4 vertically separated blocks (named July, August, September and October) and leaving only the days of the month as X-axis labels.
Figure 9: The % on the black portion of the pie-chart is not readable.
406-412: At the end of the paper, I recommend reversing the order of the sentences in this way: << In this work it was not possible to obtain a sufficiently large sample ... However, the characteristics identified with ...>.
424-482: Please, check the homogeneity in the format of the references.
Reviewer 3 Report
This study aimed to investigate the habits of public transportation users in Slovenia. A new way to travel is now available in Slovenia, based on data from tickets purchased and validated afterward. This allows for better mid- and long-term planning.
The results demonstrated that Intelligent transportation system technology (in this research, an e-ticketing system with check-in only at the time of boarding) could satisfactorily fulfill the planning and management problem outlined.
The topic of this paper is interesting, where public transportation in the regional area is often inefficient. A deeper investigation into this topic could better respond to short- and long-term changes (daily, weekly, and monthly) produced by extraordinary conditions and events in travel demand. However, I would recommend modifying the following points:
- Abstract: Line-12: off-peak instead of-peak.
- There is no need to mention "from here on" at the beginning of any abbreviation; it is sufficient to put the abbreviation in parentheses (---). This should be fixed everywhere in the manuscript.
- NFC (Line 78) should be defined.
- (65+) Line 118 and (25-59) Line 124 should add “years old” to indicate the age.
- It is great to perform some statistical analysis that supports the discussion and shows the strengths of the research, in addition to some comparisons with previous studies in Slovenia and other countries around the world.
Round 2
Reviewer 1 Report
Thank the authors for the revision. The paper seems to be much more adequate than before. However, some previously stated problems by the reviewer remained unchanged/unsolved:
- coherency between the title and the paper: according to the reviewer's opinion, the paper does not address any kind of automated prognosis/prediction on future demand using, e.g., mathematical measures;
- the authors conclude that such an e-ticketing system can contribute to raising the efficiency of the whole PT system; meanwhile, they don't discuss the before-after efficiency of the studied PT system.
Round 3
Reviewer 1 Report
Thank the authors for the revision, I agree with the changes.